# Phytochemical Composition, Antibacterial, Antioxidant and Antidiabetic Potentials of *Cydonia oblonga* Bark

**DOI:** 10.3390/molecules27196360

**Published:** 2022-09-26

**Authors:** Shaymaa Najm Abed, Sania Bibi, Marwa Jan, Muhammad Talha, Noor Ul Islam, Muhammad Zahoor, Fakhria A. Al-Joufi

**Affiliations:** 1Nursing Department, College of Applied Medical Sciences, Jouf University, Sakaka 42421, Saudi Arabia; 2Department of Biochemistry, University of Malakand, Chakdara 18800, Pakistan; 3Department of Microbiology, University of Swabi, Swabi 23562, Pakistan; 4Department of Chemistry, University of Malakand, Chakdara 18800, Pakistan; 5Department of Pharmacology, College of Pharmacy, Jouf University, Aljouf 72341, Saudi Arabia

**Keywords:** HPLC analysis, GC-MS analysis, antioxidant activity, antibacterial activity

## Abstract

*Cydonia oblonga* is a medicinal plant that is used to treat a number of health complications in traditional medication systems. The objective of this study was to evaluate the phytochemical composition, and antibacterial, antioxidant, and ant-diabetic potentials of methanolic extracts of *Cydonia oblonga* bark. The *Cydonia oblonga* bark extraction was fractionated through HPLC and seven purified fractions labeled as F1, F2, F3, F4, F5, F6, and F7 were obtained. The HPLC-UV analysis of methanolic extract showed the presence of a number of possible compounds. The GC-MS and HPLC analysis confirmed the presence of the following bioactive compounds in the crude extract and purified fractions: malic acid, mandelic acid, quercetin, caffeic acid, catechin hydrate, as morin (HPLC analysis), BIS-(2-ethylhexyl)phthalate and diisooctyl phthalate (F1), carbamide (F2, used as fertilizer), octasiloxane and dimethylsiloxanecyclictrimer (F3), silicic acid and cyclotrisiloxane (F4), 6-AH-cAMP, 4H-cyclopropa[5′,6′]benz[1′,2′,7,8]azule, and 4-(4-chlorophenyl)-3-morpholinepyrol-2-yl)-butenedioic acid (F5), isopropyamine (F6), and 1-propylhydrazine (F7). The extract and purified fractions were then tested for biological activities. All the purified fractions and methanolic extract showed effective antibacterial activity; however, the highest activity was recorded for methanolic extract against *Staphylococcus aureus* and *Streptococcus pneumonia*. Antioxidant evaluation of methanolic extract and purified fractions against DPPH showed strong % inhibition of the synthetic free radical. The methanolic extract exhibited 87.41 ± 0.54% inhibition whereas fractions showed: F1, 85.45 ± 0.85; F2, 65.78 ± 0.68; F3, 58.61 ± 0.58; F4, 80.76 ± 0.59; F5, 571.29 ± 0.49; F6, 85.28 ± 0.94; and F7, 48.45 ± 0.62% inhibition. Ascorbic acid (standard) was used as a control with 94.88 ± 0.56% inhibition at a maximum concentration of 1000 µg/mL. The α-glucosidase inhibition assay of methanolic extract and purified fractions at a maximum concentration of 1000 µg/mL showed activities as: methanolic extract, 78.21 ± 0.67; F1, 55.01 ± 0.29; F2, 56.10 ± 0.24; F3, 62.44 ± 1.03; F4, 70.52 ± 0.15; F5, 62.18 ± 0.92; F6, 72.68 ± 0.2; and F7, 57.33 ± 0.05% inhibition. α-Amylase % inhibition of methanolic extract and purified fractions were noted as: methanolic extract, 77.98 ± 0.57; F1, 79.72 ± 0.02; F2, 79.72 ± 0.02; F3, 82.16 ± 0.48; F4, 77.37 ± 0.28; F5, 72.14 ± 0.30; F6, 74.24 ± 0.29; and F7, 56.58 ± 0.10 at the highest concentration of 1000 µg/mL. Acarbose (standard) showed 87.65 ± 0.71% inhibition of α-glucosidase and 85.99 ± 0.44% inhibition of α-amylase at the highest concentration of 1000 µg/mL. It was found that all biological activities of methanolic extract and purified fractions might be attributed to the fact that they are rich sources of phenolic and flavonoids along with other bioactive compounds. The total phenolic and flavonoid contents of methanolic extract were recorded higher as compared to purified fractions (TPC = 70% and TFC = 69%). Amongst the purified fractions, fraction 6 exhibited the highest TPC value (64%), and purified fraction 1 exhibited the highest value of TFC (58%). Recent research demonstrated that *Cydonia oblonga* may be considered an antibacterial medicinal plant. The result of the present study revealed that it might be utilized for the isolation of bioactive phytochemicals that can lead to new opportunities in the discovery of new antibiotics.

## 1. Introduction

Much work has been carried out on studying and isolating natural compounds with pharmacological, pesticidal, and dietary benefits. Natural products have attracted a lot of attention in research. Since the earliest days, natural products have been employed as therapeutic agents for curing diseases [1]. The WHO estimates that more than 80% of the world’s population uses herbal medicines [2]. Herbal medicine is still incorporated into the basic healthcare system despite being replaced by conventional pharmaceuticals. The potential health and pharmacological benefits of phytochemicals have been highlighted, including their antioxidant, antibacterial, anti-cancer, cardioprotective, immune system-boosting, UV radiation-protective [3], and anti-inflammatory [3,4] properties. Additionally, medicinal plants are thought to be an important source in the development of new drugs. Synthetic medicines are constantly replacing effective pharmacological phytochemicals. However, the value of natural products cannot be emphasized enough, as herbal medicine offers a wide range of therapeutic benefits and may be used to treat a wide range of illnesses [5], including chronic diseases such as diabetes and related health complications of oxidative stress. Diabetes is considered to be the third most life-threatening health complication after cancer and heart attacks. Oxidative stress is caused by a number of causative agents and leads to number of health issues including diabetes. The most important causative agent of oxidative stress in the human body is the production of free radicals, also called reactive oxygen species, which cause the breaking of biologically important molecules such as protein and DNA, thus leading to a number of health issues. They are produced in normal oxidation processes in the body. Phenolic compounds, by virtue of a benzene ring, can stabilize these free radicals, which is why vegetables and fruits are important to be ingested in a person’s diet [1,2,3,4,5].

Secondary metabolites are important components of the human diet as they are present in daily food products such as coffee, tea, soybean, fruits, and vegetables [6]. Due to their advantages in human health, including the treatment and prevention of many illnesses, phenolic compounds and flavonoids are well known as antioxidants and other significant bioactive agents. Various studies have revealed that the consumption of polyphenols has a protective effect against inflammation and cancer [7]. Because they are known to scavenge and stop the generation of reactive oxygen and nitrogen species, the anti-inflammatory actions of polyphenols have been attributed exclusively to their antioxidant activity [8]. Moreover, phenolic-rich plants have anti-tumor [9], anti-mutagenic, anti-HIV [10], and anti-diabetic [11] properties.

Antimicrobial resistance has caused serious concern for both human health and the control of bacterial infectious illnesses [12]. Due to the intensive misuse of antibiotics to treat infections, bacterial resistance has increased in recent years [13]. A number of common pathogenic strains already bear antibiotic-resistant genes, and, presumably, more antibiotic-resistant pathogens will emerge in the future, if suitable measures are not taken to prevent this [14]. Antimicrobial resistance may be handled in two different ways, either by developing new active materials, which has growing bench-to-market costs and is time consuming [15], or by looking for ways to make existing antimicrobials more effective [16]. As phytochemicals have antibacterial potential [3], they may be potential candidates to counter antibacterial resistance. Diabetes mellitus is a metabolic illness with numerous etiologies marked by a lack of glucose homeostasis as well as abnormalities in carbohydrate, lipid, and protein metabolism as a result of deficiencies in insulin production and/or action [17]. Raised blood glucose is one of the most significant public health issues in the world; after high blood pressure and cigarette smoking, it is the third most significant risk factor for early mortality globally, according to a study by the International Diabetes Federation [18]. Well-known anti-inflammatory phytochemicals such as flavonoids, polyphenols, and tannins, which function as free radical scavengers, may have anti-diabetic effects [19,20]. These antioxidants are considered to have an insulin-mimetic impact on peripheral tissues, either by promoting tissue regeneration or by causing existing cells to secrete pancreatic insulin. The phytochemicals may also speed up filtration and renal excretion, improving metabolism, or integrating glucose into fat deposits, which improves the efficiency of the pancreas to make insulin [19].

Quince (*Cydonia oblonga*) belongs to the family Rosaceae, which is conventionally popular for medicinal, nutritional, and ornamental uses [21,22]. The name of the Cydonia genus is derived from the name of a region Kydonia on the northwestern Coast of Crete, Greece, where this tree has been cultivated since ancient times. In older Greek rituals, the fruit was offered at weddings as they symbolized fertility [23]. The fruits of quince are used in various countries for food, pharmaceutical, perfume, and agricultural purposes [24]. Therefore, this plant has garnered considerable attention due to its economic value. Secondary metabolites (phenolic compounds and essential oils), minerals, and vitamins are found in large amounts in quince fruits [25]. Various studies have revealed that the pulp of quince reduces the risk of diseases such as cardiovascular diseases [26], cancer [27], colon issues, and kidney problems [28]. Quince tannins are used as an antiseptic agent in pulmonary infections as they show activity against bacteria, fungi, and viruses [29]. Furthermore, the seed of quince is used for the treatment of cough, diarrhea, constipation, dysentery, and bronchitis. Numerous secondary metabolites have been identified in the phytochemical screening of quince fruits [30].

The bark of the *Cydonia oblonga* plant has not been studied to date. Therefore, the aim of the present work was to determine and characterize of the secondary metabolites of *Cydonia oblonga* bark using high-performance liquid chromatography, and to investigate the antioxidant, antibacterial, and anti-diabetic activity of the bark extract and the seven isolated fractions.

## 2. Results

### 2.1. Phytochemical Evaluation

Preliminary phytochemical screening of Met + Ext of *Cydonia oblonga* showed positive results for major phytochemical constituents as given in Table 1.

### 2.2. TFC and TPC in the Met-Ext and Purified Fractions of Cydonia oblonga

The results of TPC and TFC are presented in Figure 1. Met-Ext has the highest TFC and TPC values of 69% and 70%, respectively, while purified fraction 6 has the highest TPC (64%) and purified F1 has the highest TFC (58%).

### 2.3. HPLC Analysis of Methanolic Extract of Cydonia oblonga

The human health benefits of phenolic compounds are widely documented. HPLC analysis was carried out to determine the phenolic content of the Met + Ext of *Cydonia oblonga* bark. Figure 2 displays the extract chromatogram and Table 2 displays the possible compounds. Although there are many peaks in the chromatogram, only a few were identified by comparing their retention times with that of available standards. Furthermore, it should be noted that they are only possible compounds as HPLC is not an absolute technique for compound identifications.

### 2.4. GC-MS Characterization of Methanolic Extract and Purified Fractions

The results of GC-MS analysis of the fractions are given in the Supporting Materials in Appendix A (Appendix A are the GC chromatograms, Appendix A represents structure of compounds identified whereas Appendix A represents the mass fragmentation of individual compounds identified in each fraction) whereas Appendix A correspondingly summarizes the major compounds present in fractions F1 to F7. The identified compounds are depicted in Table 3.

### 2.5. Evaluation of Antibacterial Activity of Methanolic Extract and Fractions

Using the disc diffusion methodology, the antibacterial activities of the Met + Ext and purified fractions were assessed against several microbial strains. The zones of inhibition were noted and the results are presented in Table 4.

### 2.6. Antioxidant Activity of Crude and Purified Fractions of Cydonia oblonga

DPPH scavenging activity was investigated to find the antioxidant potential of the Met + Ext and the purified fractions and results are displayed in Table 5. The standard ascorbic acid caused scavenging up to 94.88 ± 0.96% with the lowest IC_50_ value of 30 µg/mL at 1000 µg/mL against DPPH. Met + Ext exhibited 87.41 ± 0.54 with the lowest IC_50_ value of 120, while among the fractions the highest % inhibition value was recorded for F1, as 85.45 ± 0.85, with the lowest IC_50_ value of 115.

### 2.7. In Vitro Antidiabetic Potential of Met + Ext and Purified Fractions

#### 2.7.1. In Vitro α-Glucosidase Enzyme Inhibition Activity

The IC_50_ values of the purified fractions and Met + Ext were calculated at different concentrations as presented in Table 6. Met + Ext was the most potent inhibitor with the lowest IC_50_ value. Acarbose (standard) at the maximum concentration 1000 µg/mL showed 87.65 ± 0.71 α-glucosidase inhibition with 30 µg/mL IC_50_ value.

#### 2.7.2. In Vitro α Amylase Enzyme Inhibition Activity

The IC_50_ values of the purified fractions and Met + Ext were calculated at different concentrations along with percent inhibition values and are depicted in Table 6. The crude extract was recorded to be the most potent α-amylase inhibitor l with the lowest IC_50_ value. Standard acarbose at the maximum concentration 1000 µg/mL showed 85.99 ± 0.44 α-glucosidase inhibition with 32 µg/mL IC_50_ values.

## 3. Discussion

The analysis of the present study indicated that Met-Ext had the highest TFC and TPC values of 69% and 70%, respectively. Purified fraction 6 had the highest TPC of 64% and purified fraction 1 had the highest TFC of 58%. According to Erdoğan et al. [25], the leaves of *Cydonia oblonga* are a rich source of phenolic compounds, which confirms our observed results. Scientific interest in the standardization and characterization of herbal medications is ongoing in the herbal drug sector. Simple, rapid, practical, and affordable standardization approaches are becoming more and more desirable as sophisticated chromatographic techniques are introduced [27]. For the standardization of Met + Ext from plant bark, HPLC was used as it is a sensitive and accurate technique that is widely used for the evaluation of plant extracts and their derived products/formulations [31]. The HPLC-UV analysis of Met + Ext showed the presence of a number of possible compounds such as malic acid, mandelic acid, quercetin, morin, catechin hydrate, and caffeic acid, which were identified through comparison of the retention time of the respective peak in the chromatogram with that of the available standards. As mentioned in the Section 2, this is not a standard technique for the identification of compounds and thus must be used in conjunction with MS in the form of GC-MS or LC-MS and/or NMR investigations in order to fully identify the compounds present and comprehend their structure [32]. The GC-MS technique was successfully employed for *Cydonia oblonga* bark isolated fractions, and is one of the most extensively used techniques for separating phytoconstituents. About 25 phytochemical substances in the leaves and rhizome of *Amomum nilgiricum* species have been identified previously through GC-MS [33]. The GC-MS analysis of the purified fractions also confirmed the presence of certain valuable phytochemicals. F1 contains bis(2-ethylhexyl) phthalate and diisooctyl phthalate with a retention time of 22.90 min. Carbamide was found in F2 with a retention time of 22.2 min. Octasiloxane and dimethylsiloxanecyclictrimer were confirmed in F3 with their retention time of 25.57 min. Silicic acid and cyclotrisiloxane with retention times of 25.67 min were found in F4. 6-AH-cAMP, 4H-cyclopropa[5′,6′]benz[1′,2′.7,8]azure 4-(4-chlorphenyl)-3-morpholino-pyrrol-2-yl)-butenedioic acid were active compounds of F5 with retention times of 25.61 min. Isopropylamine was found in F6 with a retention time of 2.30 min. 1-Propylhydrazine was found as an active compound in F7 with a retention time of 1.04 min. Being purified fractions, few bioactive compounds were present in each fraction.

Antibiotics are widely used to address a variety of infections brought on by bacteria, which can be multidrug resistant. These infections are common, which raises treatment resistance and impedes the research and development of new drugs [34]. Analysis has revealed that Gram-positive bacteria are more vulnerable than Gram-negative bacteria to the tested extract and fractions. Our research is consistent with the reported studies where it has been discovered that quince extract has a significant antibacterial effect against Gram-positive bacteria but is less effective against Gram-negative bacteria [35,36]. The outer membranes of Gram-negative bacteria contain lipopolysaccharide, which keeps phenolics from adhering to the surface of the bacterial strain [37], which is why lower activity has been observed against each Gram-negative strain. In general, Gram-positive bacteria appear to be more sensitive to bactericidal polyphenols than Gram-negative species [38]. The DPPH scavenging activity of methanolic and the purified fractions were measured, and the results exhibited potential antioxidant activity. The leaf extract of quince has depicted a concentration-dependent antioxidant potential of 120 µg/mL [39] previously. In another study, methanolic extracts of edible peel and pulp of quince showed IC_50_ values of 600 and 800 µg/mL, respectively, whereas seed extracts were reported to have a lower antioxidant potential [40]. The *Cydonia obolonga* extract and the isolated fractions showed effectiveness against α-glucosidase and α-amylase. A previous study showed that plant extracts (methanolic and n-hexane) display remarkable anti-diabetic activity against α-glucosidase and α-amylase [41]. The antidiabetic activity of *Cydonia oblonga* leaves in hydro-ethanolic extract was studied in normal and streptozotocin-induced diabetic rats. There was no significant effect on normal rat glucose, while a significant reduction in blood glucose levels was recorded in diabetic rats at the time period of 0 to 3 h [42,43]. These reported findings confirm our observations on the selected plant.

## 4. Material and Methods

### 4.1. Plant Sample Collection

The bark of *Cydonia oblonga* was collected from Allahdand Dheri (latitude. 34.61012° or 34°36′3″ north; longitude. 72.0187° or 72°1′7″ east), Malakand Khyber Pakhtunkhwa (KP), Pakistan. The taxonomic identity of the plant was authenticated by a botanist in the Department of the Botany University of Swabi.

### 4.2. Chemicals and Standards

Gallic acid and quercetin were purchased from Sigma-Aldrich France, and nutrient agar was obtained from Oxoid Ltd. England. 2,2-Diphenyle-1 picrylhydrazyl (DPPH) and Folin–Ciocalteu reagent (Sigma-Aldrich CHEMIE GmbH Burlington, MA, USA), aluminum chloride, methanol, iron chloride, sodium hydroxide, sodium nitrite, and ascorbic acid were purchased from Sigma-Aldrich, Germany. All the chemicals utilized in assays were of analytical grade, except HPLC-grade solvents, which were purchased from Dae-Jung, Korea. They were used as such without any further purification.

### 4.3. Extraction and Fractionation

The fresh bark of the *Cydonia oblonga* was washed and dried under a hood to avoid deterioration of bioactive active compounds due to sunlight. The dried sample was ground into a powder and mixed well with methanol. After three days, the extract was filtered through Whatman filter paper, and using a rotary evaporator (Rota Vapor R-200 Buchi, Switzerland) the solvent was evaporated from the filtrate under reduced pressure and at 40 °C. HPLC analysis of the crude was carried out using a reported protocol [44]. Then, 7 HPLC fractions were collected in vials which were labeled as F1, F2, F3, F4, F5, F6, and F7 and stored at 4 °C temperature till further analysis. The separated fractions and methanolic extract (Met + Ext) were screened for phytochemical evaluation (TPC and TFC etc.) and various biological activities. The purified fractions were also analyzed using GC-MS for the presence of bioactive compounds.

#### 4.3.1. Phytochemical Analysis

Preliminary phytochemical screening was undertaken to determine the presence of various bioactive constituents such as alkaloids, flavonoids, glycosides, terpenoids, and tannins in Met + Ext of *Cydonia oblonga* bark using reported assays [45,46,47].

#### 4.3.2. Analysis of Total Phenolic Content and Total Flavonoids Content

The content of phenolics was determined in *Cydonia oblonga* bark Met + Ext and purified fractions from F1 to F7 using Folin–Ciocalteu assay [45]. For the preparation of the required solution, 5 mg of sample extract was taken and dissolved in 5 mL of methanol. Then, 1 mL of Folin–Ciocalteu reagent was diluted with 10 mL of distilled water to prepare the Folin–Ciocalteu reagent solution. In a volumetric flask, 9 mL of distilled water was added to 1 mL of sample solution and 1 mL of the diluted solution of Folin–Ciocalteu reagent and allowed to stand for 6 min. After 6 min, 10 mL of 7% NaHCO_3_ (sodium carbonate) solution was successively added to the mixture. The mixture was made up to 25 mL volume with distilled water and thoroughly mixed. After 90 min, the absorbance was measured at 760 nm using a UV spectrophotometer. Gallic acid solution curve (0 to 100 mg/mL) was used as a standard. Total phenolic content was expressed as gallic acid equivalents (mg GAE/g) per gram of dry sample, which is a common reference compound.

The total flavonoid content was determined using the aluminum chloride calorimetry method [46] with little modification. The plant extract was diluted with methanol. The methanolic solution of extract (100 µL) and 500 µL distilled water were added to 10 mL volumetric flasks followed by the addition of 100 µL NaNO_2_ (5%), 150 µL AlCl_3_ (10%), and 200 µL NaOH (1 M). A calibration curve was prepared by using the methanolic solution of quercetin in concentrations ranging from 0–100 μg/mL. The resulting mixture was incubated for 5 min; absorbance was measured spectrophotometrically at 510 nm. The total flavonoid content was expressed in terms of Quercetin equivalent to mg of QE/g of dry samples.

#### 4.3.3. HPLC Characterization

##### Sample Preparation for HPLC Analysis

HPLC analysis was performed according to a previously reported protocol [44] with some modifications. For analysis, a sample extract was prepared by dissolving 1 g of powder sample in a 10 mL mixture of water and methanol (5:5). The mixture was shaken thoroughly for 1 h at 60 °C and then subjected to filtration using a Whatman filter paper with 0.7 µm pore size. The filtrates were then centrifuged for 15 min at 4000 rpm. The sample extract was filtered twice and collected in a HPLC vial, marked with proper code.

##### HPLC-UV

The sample was analyzed in an Agilent 1260 infinity High-performance Liquid Chromatography (HPLC) instrument equipped with an autosampler, quaternary pump, degasser, and ultraviolet (UV) detector. An Agilent Zorbax Eclipse C18 column was used for separation. The mobile phase was solvent A (acetic acid, methanol, deionized water, 2.10.88, *v*/*v*) and solvent B (acetic acid, deionized water, methanol, 2.8.90, *v*/*v*). The system was conditioned with the mobile phase at least for 30–40 min. The gradient profile started with 100% solvent A at 0 min, 85% A at 5 min, 50% solvent A at 20 min, at 25 min solvent A decreased to 30%, and finally, 100% solvent B at 30–40 min. The solvents were delivered at 1 mL/min flow rate, and the absorbance of phytochemicals was measured at 280 nm wavelength, by using retention times (chromatographic comparisons), and UV spectra were recorded with the UV detector identification of different compounds. Seven active fractions were obtained (F1 to F7), which were further analyzed by GC-MS.

##### GC-MS Analysis for Characterization

Phytoconstituents in purified fractions were analyzed through GC-MS (Agilent Technological, Santa Clara, CA, USA) using a previously reported protocol [48]. The system was equipped with an FID detector to identify phytochemicals. Compounds were identified by comparing their mass spectra and retention times with pure compounds (standards) taken from Willy and NIST libraries software.

### 4.4. Antibacterial Activity

In vitro antibacterial activity was examined for Met + Ext and purified fractions of *Cydonia oblonga* against Gram-positive (*Bacillus subtilis*, *Staphylococcus aureus*, and *Streptococcus pneumonia*) and Gram-negative bacteria (*Escherichia coli*, *Salmonella typhi*, and *Klebsiella pneumonia*) using the disc diffusion method. Standard bacterial cultures were used in which the viable count in each culture was determined using a surface viable technique. The bacterial cultures (from 10^8^–10^9^ colony forming units per mL) were swabbed uniformly on the surface of the prepared media using a separate sterile cotton swab each time. Sterilized filter paper discs (Whatman No. 2, Whatman plc, Little Chalfont, GB) impregnated with 20 µL of sample solution were placed individually at equal distances on the surface of seeded agar plates, followed by incubation of bacteria at 37 °C for 24 h in an incubator. Respective solvents without plant extracts were taken as a negative control. After incubation, the results were observed, and we measured the zone diameter around each disc. The experiment was performed in triplicate and mean zone of inhibition values were calculated.

### 4.5. DPPH Radical Scavenging Activity

The stock solution was prepared by dissolving 20 mg DPPH in 100 mL of methanol; 3 mL of this solution was taken and its absorbance was measured spectrophotometrically at 517 nm and was considered as a negative control solution. For free radical formation, a stock solution was covered with aluminum foil and incubated at room temperature in dark conditions for 24 h. About 5 mg of extract and each purified fraction were dissolved in 5 mL methanol to obtain the stock solution. Then, the above solution was serially diluted in methanol at the following concentrations: 1000, 500, 250, 125, and 62.5 (µg/mL). Then, 2 mL of each sample solution was mixed with 2 mL solution of DPPH and after reaction for 15 min in dark, the absorbance values of the samples were measured. DPPH radical scavenging activity of plant extract and fractions were calculated by using the following equation:(1)$$\%\;Scavanging\;activity=\frac{A○-B○}{A○}\times 100\;$$
where *A*○ is the absorbance of pure DPPH and *B*○ is the absorbance of samples. IC_50_ values were determined using ascorbic acid taken as a standard, prepared with a concentration of 5 mg/5 mL. Serial dilutions were performed with different concentrations of ascorbic acid in the samples at 1000, 500, 250, 125, and 62.5 µg/mL. All the experiments were performed in triplicate.

### 4.6. Antidiabetic Assays

#### 4.6.1. Inhibition of α-Amylase

The α-amylase solution was prepared by dissolving 10 mg enzyme in 100 mL distilled water. Then, 10 µL α-amylase enzyme solution was mixed with each test sample in the concentration range of 30 µL. As a negative control, 30 mL of sodium phosphate buffer was added to 10 µL enzyme solution and incubated at 37 °C for 10 min. Then, 40 mL of starch solution was added to each test tube. Incubation was carried out again at 37 °C for 30 min, then 20 µL of 1 M HCL was added and absorbance was measured at 540 nm using a spectrophotometer. The control samples were prepared without any plant samples. Equation (2) was used to estimate the percent inhibition of the enzyme.

#### 4.6.2. α-Glucosidase Inhibition Activity

The reaction mixture was formulated by adding 100 µL of α-glucosidase enzyme (0.5 units/mL), 0.1 M phosphate buffer (600 µL) at pH 6.9 and 50 µL of each sample dilution (62.4, 125, 250, 500, and 1000 µg/mL). The mixture was incubated for 15 min at 37 °C. The enzymatic reaction began by adding 100 µL *p*-nitro-phenyl-α-D-glucopyranoside (5 Mm) solution in 0.1 M phosphate buffer at pH 6.9 followed by 15 min incubation at 37 °C. The absorbance of the final reaction mixture was recorded at 405 nm. The reaction mixture with no plant extract was used as positive control while the blank solution was prepared without the enzyme α-glucosidase. The α-glucosidase % inhibition was calculated using the formula:(2)$$\%\;\;\alpha -glucosidase\;inhibition=\frac{control\;absorbance-sample\;absorbance}{control\;absorbance}\times 100$$

### 4.7. Statistical Analysis

To calculate IC_50_ for % DPPH, α-amylase, and α-glucosidase inhibition, Excel 2007 was used. The standard deviation and means were estimated using Excel.

## 5. Conclusions

HPLC analysis revealed that the bark of *Cydonia oblonga* has numerous biologically active compounds such as polyphenols and flavonoids. The crude extract and fractions isolated through HPLC were subjected to GC-MS analysis. A total of twelve compounds were identified through HPLC and GC-MS analysis. The extract and various fractions exhibited potential biological activity such as antioxidant, antibacterial, and anti-diabetic activities. This bark can constitute an alternative therapeutic agent for bacterial infections, oxidative stress, and diabetes, which are major subjects for future research. Pharmacological and toxicological studies are also needed to confirm the hypothesis tested here, which will also further elaborate on the safety profile of the extract.

## Figures and Tables

**Figure 1 molecules-27-06360-f001:**
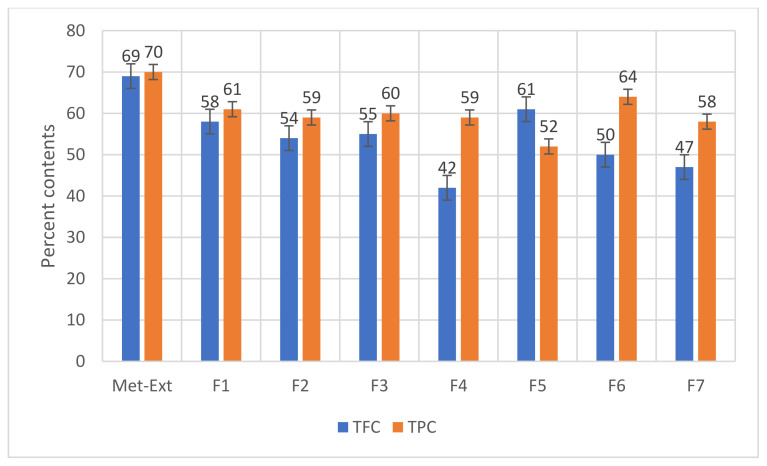
TPC and TFC of Met-Ext and purified fractions of *Cydonia oblonga*.

**Figure 2 molecules-27-06360-f002:**
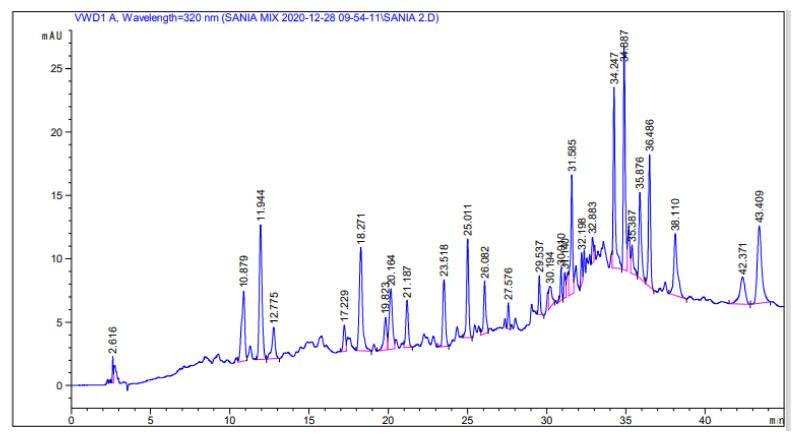
HPLC-UV chromatogram of bioactive compounds of the *Cydonia oblonga* Met + Ext.

**Table 1 molecules-27-06360-t001:** Preliminary qualitative phytochemicals screening of *Cydonia oblonga* Met + Ext.

Phytochemicals	Reagent	Analyses	Result
Flavonoids	Ferric chloride	Appearance of yellow color and colorless after HCL addition	+
Alkaloids	Dragendroff’s	Formation of orange–red color precipitate	+
Glycosides	Keller Killiani	Red to brown layer formation	+
Triterpenoids	Liebermann Burchard	Reddish–brown boundary	+
Tannins	Gelatine	Brownish to green precipitates	+

The + sign represents the presence of a given phytochemical group.

**Table 2 molecules-27-06360-t002:** Identification of bioactive compounds in Met + Ext of *Cydonia oblonga* through HPLC-UV technique.

Retention Time (min)	Phenolic Compound Identity	Peak Area of Sample	Identification Reference
2.616	Malic acid	7.64427	Ref. Stand
10.879	Mandelic acid	28.98360	Ref. Stand
12.373	Caffeic acid	21.41109	Ref. Stand
20.578	Quercetin	77.68702	Ref. Stand
29.537	Catechin hydrate	46.64664	Ref. Stand
30.408	Morin	134.41394	Ref. Stand

**Table 3 molecules-27-06360-t003:** List of the identified compounds in *Cydonia oblonga bark* fractions (F1-F7) determined through GC-MS analysis.

S. No	Fraction	Compounds
1	F1	BIS-(2-ethylhexyl)phtjalate, diisooctyl phthlate
2	F2	carbamide
3	F3	octasiloxane, dimethylsiloxanecyclictrimer
4	F4	silicic acid, cyclotrisiloxane
5	F5	6-AH-cAMP, 4H-cyclopropa[5′,6′]benz[1′,2′,7,8]azule, 4-(4-chlorophenyl)-3-morpholinepyrol-2-yl)-butenedioic acid
6	F6	isopropyamine
7	F7	1-propylhydrazine

**Table 4 molecules-27-06360-t004:** Antibacterial activity of Met + Ext and purified fractions of *Cydonia oblonga* using agar well diffusion method.

Sample	Zone of Inhibition (mm)					
	Microbial Strains					
	*Escherichia coli*	*Salmonella typhi*	*Klebsiella pneumonia*	*Bacillus subtilis*	*Staphylococcus aureus*	*Streptococcus pneumonia*
**Crude**	18 ± 0.7	19 ± 2.0	18 ± 0.6	19 ± 0.2	25 ± 1.8	20 ± 2.2
F1	10 ± 0.6	10 ± 0.9	10 ± 0.8	10 ± 0.8	15± 0.4	12 ± 1.1
F2	11 ± 0.2	14 ± 0.4	10 ± 1.3	15 ± 1.5	15 ± 1.1	20 ± 2.0
F3	11 ± 1.9	10 ± 1.3	15 ± 0.9	10 ± 1.2	10 ± 1.2	16 ± 1.6
F4	15 ± 1.4	13 ± 0.8	11 ± 1.0	12 ± 1.5	11 ± 0.5	13 ± 1.4
F5	9 ± 1.6	10 ± 1.6	12 ± 0.6	16 ± 1.0	14 ± 0.3	11 ± 0.8
F6	9 ± 08	12 ± 1.2	12 ± 1.9	10 ± 1.1	9 ± 1.7	11 ± 1.3
F7	10 ± 0.4	9 ± 0.5	12 ± 1.3	11 ± 0.8	10 ± 1.3	12 ± 1.4

**Table 5 molecules-27-06360-t005:** DPPH free radical scavenging potential of *Cydonia oblonga* at various concentrations.

S. No	Sample	Concentration (µg/mL)	% DPPH Scavenging	IC_50_
1	Met-Ext	1000	87.41 ± 0.54	120
		500	85.12 ± 0.76	
		250	68.27 ± 0.85	
		125	52.78 ± 0.25	
		62.5	36.98 ± 0.63	
2	F1	1000	85.45 ± 0.85	115
		500	74.23 ± 0.75	
		250	67.12 ± 0.91	
		125	57.34 ± 0.77	
		62.5	38.65 ± 0.48	
3	F2	1000	65.78 ± 0.68	200
		500	60.27 ± 0.63	
		250	57.92 ± 0.78	
		125	46.74 ± 0.89	
		62.5	36.04 ± 0.43	
4	F3	1000	58.61 ± 0.58	380
		500	52.05 ± 0.49	
		250	48.93 ± 0.75	
		125	44.21 ± 0.48	
		62.5	34.96 ± 0.81	
5	F4	1000	80.76 ± 0.59	110
		500	68.23 ± 0.67	
		250	64.04 ± 0.83	
		125	54.92 ± 0.79	
		62.5	42.71 ± 0.58	
6	F5	1000	71.29 ± 0.49	200
		500	67.61 ± 0.59	
		250	57.93 ± 0.59	
		125	46.97 ± 0.59	
		62.5	37.12 ± 0.69	
7	F6	1000	85.28 ± 0.94	105
		500	81.60 ± 0.96	
		250	70.26 ± 0.86	
		125	55.82 ± 0.89	
		62.5	36.05 ± 0.99	
8	F7	1000	48.45 ± 0.62	110
		500	40.38 ± 0.38	
		250	34.18 ± 0.85	
		125	31.41 ± 0.45	
		62.5	29.34 ± 0.63	
10	Ascorbic acid	1000	94.88 ± 0.56	30
		500	86.59 ± 0.45	
		250	78.64 ± 0.76	
		125	69.14 ± 0.25	
		62.5	62.87 ± 0.53	

**Table 6 molecules-27-06360-t006:** α-Glucosidase and α-amylase inhibitions of *Cydonia oblonga* of methanolic extract and subsequent purified fractions at different concentrations.

Sample	Concentration (µg/mL)	% α-Glucosidase Inhibition	IC_50_ (µg/mL)	% α-Amylase Inhibition	IC_50_ (µg/mL)
		Mean ± SEM		Mean ± SEM	
Met-Ext	1000	78.21 ± 0.67	57	77.98 ± 0.57	52
	500	73.51 ± 0.56		76.29 ± 0.36	
	250	64.79 ± 0.14		74.32 ± 0.25	
	125	60.28 ± 0.38		68.38 ± 0.68	
	62.5	52.51 ± 1.06		57.28 ± 0.52	
F1	1000	55.01 ± 0.29	500	79.72 ± 0.02	57
	500	50.97 ± 0.39		78.22 ± 0.72	
	250	43.13 ± 0.34		74.02 ± 0.76	
	125	47.24 ± 0.29		67.97 ± 0.80	
	62.5	32.41 ± 0.74		52.58 ± 0.10	
F 2	1000	56.10 ± 0.24	494	79.97 ± 0.30	45
	500	51.88 ± 0.38		79.27 ± 0.28	
	250	47.62 ± 0.91		76.22 ± 0.62	
	125	47.09 ± 0.01		75.82 ± 0.56	
	62.5	30.89 ± 1.99		70.18 ± 0.48	
F3	1000	62.44 ± 1.02	200	82.16 ± 0.48	49
	500	57.61 ± 0.28		80.41 ± 0.92	
	250	55.88 ± 0.18		78.06 ± 0.65	
	125	48.19 ± 0.15		76.72 ± 0.53	
	62.5	43.27 ± 1.02		74.67 ± 0.59	
F4	1000	70.52 ± 0.15	370	77.37 ± 0.28	54
	500	63.86 ± 0.03		76.57 ± 0.04	
	250	55.04 ± 0.08		74.47 ± 0.92	
	125	49.04 ± 0.71		68.88 ± 0.38	
	62.5	43.25 ± 0.90		57.18 ± 0.05	
F5	1000	62.18 ± 0.92	180	72.14 ± 0.30	220
	500	59.92 ± 0.49		61.50 ± 0.58	
	250	52.92 ± 0.32		53.44 ± 0.28	
	125	48.84 ± 0.73		46.16 ± 0.74	
	62.5	36.73 ± 0.07		38.58 ± 0.48	
F6	1000	72.68 ± 0.22	125	74.24 ± 0.29	59
	500	68.05 ± 0.52		67.38 ± 1.62	
	250	57.39 ± 0.99		62.67 ± 0.25	
	125	50.92 ± 0.27		56.52 ± 0.49	
	62.5	46.35 ± 0.89		51.81 ± 0.76	
F7	1000	57.33 ± 0.05	495	56.58 ± 0.10	500
	500	51.61 ± 0.12		50.42 ± 0.46	
	250	46.84 ± 0.83		44.39 ± 0.78	
	125	39.60 ± 0.39		38.28 ± 0.60	
	62.5	34.68 ± 0.06		31.09 ± 1.05	
Acarbose	1000	87.65 ± 0.71	30	85.99 ± 0.44	32
	500	83.05 ± 0.65		83.61 ± 0.58	
	250	78.90 ± 1.02		76.85 ± 0.96	
	125	71.83 ± 0.99		70.47 ± 0.78	
	62.5	65.15 ± 0.75		64.89 ± 0.71	

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
