# Peer review of "Phytochemical Composition, Antibacterial, Antioxidant and Antidiabetic Potentials of Cydonia oblonga Bark"

_molecules, 2022, doi:10.3390/molecules27196360_

Round 1

Reviewer 1 Report

My specific comment on the manuscript entitled "Phytochemical composition, antimicrobial, antioxidant and antidiabetic potentials of methanolic extract of Cydonia oblonga bark" is as follows;

1. In the introduction, authors can signify the need of novel drug candidated based on antibiotic resistance and chemotherapy resistance etc.

2. Authors mentioned about secondary metabolites in Line 69; however, they limited the content to polyphenols alone. Why it is too restricted?

3. As in Line 81, references can be given together "[10-11]. Please check throughout the manuscript

4. Initially authors used HPLC for characterization; later, they opted for GC-MS. What is the reason for the same. Needs to be explained.

5. Why IC50 values are represented as mean only? Whether authors conducted the assay for a single time? Otherwise, there would be standard deviation/ error in the values.

6. Authors missed about diabets, oxidative stress etc in discussion. Needs to be improved further.

Author Response

Reviewer 1

My specific comment on the manuscript entitled "Phytochemical composition, antimicrobial, antioxidant and antidiabetic potentials of methanolic extract of Cydonia oblonga bark" is as follows;

  1. In the introduction, authors can signify the need of novel drug candidated based on antibiotic resistance and chemotherapy resistance etc.
  • Worthy reviewer, the relevant literature was added in revised manuscript.
  1. Authors mentioned about secondary metabolites in Line 69; however, they limited the content to polyphenols alone. Why it is too restricted?
  • Worthy reviewer, as our HPLC method is specific for phenolics that is why they have been focused. Although I have made the statement generalized in the revised paper but still as you know that the main biomolecules, (poly )phenolic substances, are a class of higher plant secondary metabolites. Moreover, several studies showed Quince species have rich source of phenolic compounds. Therefore, we have the limited the content to polyphenols.
  1. As in Line 81, references can be given together "[10-11]. Please check throughout the manuscript
  • Worthy reviewer, Corrected accordingly.
  1. Initially authors used HPLC for characterization; later, they opted for GC-MS. What is the reason for the same. Needs to be explained.
  • Worthy reviewer, actually HPLC is a separating technique rather than as identification tool. The most authentic techniques are spectroscopic based techniques where for the identification of known compounds GC-MS and LC-MS are preferably used and for novel compounds NMR etc are used. HPLC is used as identification tool when standards are available and after spiking with the pure standard if a suspected peak sharply rises the compound is confirmed and if a shoulder is formed by that peak, then it is some other compound. Here we have used this spiking technique in HPLC analysis but as it is not so authentic technique therefore, its fraction has been subjected to GC-MS analysis which is an authentic technique. In short HPLC has been used for separation whereas GC-MS has been used for the identification the compounds present.
  • Also, we were interested in purified fraction rather than crude extract as crude extract contained a range of compounds for which then one is not sure which compound will be the responsible compound of the observed biological activity. If you use a purified fraction the selection become narrow and authentic.
  1. Why IC50 values are represented as mean only? Whether authors conducted the assay for a single time? Otherwise, there would be standard deviation/ error in the values.
  • Worthy reviewer, the standard deviation errors are already shown with the individual values. However, IC50 are estimated from a graph constructed for the average values of a range of concentrations which always has a single value that is the available practice in literature as well. To save our time we have constructed the graph for average rather than individual sets of experiments.
  1. Authors missed about diabets, oxidative stress etc in discussion. Needs to be improved further.
  • Discussion about anti-diabetic activity was added in revised manuscript.

Reviewer 2 Report

Minor Comments;

1.     Title needs to be improved.

2.     The anti-diabetic potential is not mentioned in the introduction and discussion part.

3.     Statistical tools used for the study are not mentioned in the manuscript

4.     The IC50 values of the anti-oxidant assay are much lower than the doses taken. Could have included lower doses for the study.

5.     Anti-oxidant and anti-diabetic data can be represented graphically for better understanding.

Abstract

Line 20- “quercetin” – check spelling.

The abstract could start with a background description of the work.

Point out major compounds in the fractions rather than every compound.

Introduction

Line 62- incorporate references for the pharmacological benefits mentioned.

Line 71- mention “other significant biological agents”

Line 72- “revealed “that,” the”- insert

Line 73- change inflammatory to inflammation.

Line 73-75 – “Because they were known to scavenge and stop the generation of reactive oxygen and nitrogen species, the anti-inflammatory actions of polyphenols have been attributed exclusively to their antioxidant activity”- check the statement as good antioxidant compounds may not have significant anti-inflammatory properties.

Line 77-78 – “In recent years, resistance developed in pathogenic microorganisms against most classes of antibiotics due to their excessive uses”- sentence lacks a proper structure.

Line 80- “Quince Cydonia oblonga plant”- change it to “Quince (Cydonia oblonga) plant”

Line 86- correct “[13]. ].”

Line 95-96 – “was the determination and characterization” modify it to “was to determine and characterize”

Line 97- “to” investigate

Line 99-100 -  delete the sentence as it is a repetition

The introduction needs to be modified as it doesn't comment on how free radicals are generated and what consequences they can have on our body and the importance of antioxidants. Further, no discussion was made on diabetics as the title itself includes “Antidiabetic Potentials”

Results

Line 103-106- could have included the % yield after extraction and % yield after fractionation

Why were the data of phenol and flavonoid content not expressed in Gallic acid equivalents (mg GAE/g) and Quercetin 315 equivalent to mg of QE/g?

Figure 1- why is the figure without SD?

Tables 5, and 6 data seem a bit confusing. It is better to represent the data as a figure

Table 6 data could have included an even lower concentration of Met+Ext and Fractions as most of the IC50 values are below the lowest concentration used (62.5 µg/mL)

Discussion

Line 166- “Shahrestani et al., 2014” citation is missing from the reference part.

Why was GC-MS used instead of HPLC-MS for the present study?

Line 188- “specie” check spelling

Line 209-211- “Met + Ext and 7 purified fractions inhibited by DPPH revealed that F4, F6, and F7 caused significant inhibition with the lowest IC50 values 110, 105, and 110, respectively, comparable to ascorbic acid”- check the sentence construction

Line 215-217- no reference cited

Why the authors have not incorporated the antidiabetic potential in the discussion which was explored in the present study?

Materials and Method

Line 221- could have added the latitude and longitude coordinates.

Line 238- the modification made to the standard protocol needed to be explained.

Line 241- it would be better to use antibacterial instead of antimicrobial as bacteria was only involved in the study.

Line 246- authors could include a note on various cultural media used

Title

Line 3- It would be better to use antibacterial potential instead of antimicrobial potential because the study mainly focuses on various bacterial groups and in conclusion the authors have pointed out the same.

Author Response

Reviewer 2

  1. Title needs to be improved.
  • The title was modified accordingly
  1. The anti-diabetic potential is not mentioned in the introduction and discussion part.
  • relevant literature was added in introduction and discussion portion.
  1. Statistical tools used for the study are not mentioned in the manuscript.
  • Section 4.7 was accordingly added
  1. The IC50 values of the anti-oxidant assay are much lower than the doses taken. Could have included lower doses for the study.
  • Worthy reviewer, in such cases extrapolation is used to estimate IC50. At present we do not have the sample to perform experiment at lower doses.
  1. Anti-oxidant and anti-diabetic data can be represented graphically for better understanding.
  • Worthy reviewer, although in thesis we have constructed these graphs but as there are at least five concentrations for each extract or fraction which means there would be 40 bars in each graph which would be somewhat cumbersome. Only for IC50 a simple graph can be constructed. If the worthy reviewer still insists then let us know we will construct it for IC50

Abstract

Line 20- “quercetin” – check spelling.

  • Worthy reviewer, spelling was corrected accordingly.

The abstract could start with a background description of the work.

  • Worthy reviewer, the abstract was updated in revised manuscript.

Point out major compounds in the fractions rather than every compound.

  • After each compound name fraction abbreviation was accordingly written in brackets.

Introduction

Line 62- incorporate references for the pharmacological benefits mentioned.

  • Worthy reviewer, the references were incorporated.

Line 72- “revealed “that,” the”- insert

  • Worthy reviewer, corrected accordingly.

Line 73- change inflammatory to inflammation.

  • Worthy reviewer, corrected accordingly.

Line 73-75 – “Because they were known to scavenge and stop the generation of reactive oxygen and nitrogen species, the anti-inflammatory actions of polyphenols have been attributed exclusively to their antioxidant activity”- check the statement as good antioxidant compounds may not have significant anti-inflammatory properties.

  • Worthy reviewer, the statement was checked thoroughly where some modification were made. However, the claim made is based on the literature findings:
  • The original statement in literature

“Several studies have shown that the high consumption of polyphenols has protective effects against cancer and inflammatory diseases (Kang, Shin, Lee, & Lee, 2011). The anti-inflammatory effects of polyphenols have been attributed primarily to their antioxidant activity because they were known to scavenge and prevent the formation of reactive oxygen and nitrogen species (Eberhardt, Lee, & Liu, 2000; Kim, Lee, Lee, & Lee, 2002; Korycka-Dahl & Richardson, 1978; Lavelli, Hippeli, Peri, & Elstner, 1999)”

Line 77-78 – “In recent years, resistance developed in pathogenic microorganisms against most classes of antibiotics due to their excessive uses”- sentence lacks a proper structure.

  • Worthy reviewer, the sentence was restructured in revised manuscript.

Line 80- “Quince Cydonia oblonga plant”- change it to “Quince (Cydonia oblonga) plant”

  • Worthy reviewer, corrected accordingly.

Line 86- correct “[13]. ].”

  • Worthy reviewer, corrected accordingly.

Line 95-96 – “was the determination and characterization” modify it to “was to determine and characterize”

  • Worthy reviewer, corrected accordingly.

Line 97- “to” investigate

  • Worthy reviewer, corrected accordingly.

Line 99-100 -  delete the sentence as it is a repetition

  • Worthy reviewer, sentence was deleted as you suggested.

The introduction needs to be modified as it doesn't comment on how free radicals are generated and what consequences they can have on our body and the importance of antioxidants. Further, no discussion was made on diabetics as the title itself includes “Antidiabetic Potentials”

  • Literatures about antioxidant and antidiabetic potentials were incorporated in introduction and discussion sections accordingly.

Results

Line 103-106- could have included the % yield after extraction and % yield after fractionation

  • It was a mistake; the section was moved to experimental section.

Why were the data of phenol and flavonoid content not expressed in Gallic acid equivalents (mg GAE/g) and Quercetin 315 equivalent to mg of QE/g?

  • Worthy reviewer, they were estimated in the mentioned units however, then they have been converted into percent. At this stage if we revise, we would have to change the related sections totally. If the worthy reviewer, still we will rearrange the sections.

Figure 1- why is the figure without SD?

  • Revised accordingly

Tables 5, and 6 data seem a bit confusing. It is better to represent the data as a figure

  • Worthy reviewer as mentioned before figures of such huge data are more confusing than the data in tables. So it would be better to retain them in the tables rather than presenting them in figures

Table 6 data could have included an even lower concentration of Met+Ext and Fractions as most of the IC50 values are below the lowest concentration used (62.5 µg/mL)

  • Worthy reviewer, at this stage we do not have the samples. Also it is estimated from extrapolation of the graph which is scientifically permissible.

Discussion

Line 166- “Shahrestani et al., 2014” citation is missing from the reference part.

  • Worthy reviewer, corrected reference was incorporated in the revised manuscript.

Why was GC-MS used instead of HPLC-MS for the present study?

  • Worthy reviewer, actually HPLC is a separating technique rather than as identification tool. The most authentic techniques are spectroscopic based techniques where for the identification of known compounds GC-MS and LC-MS are preferably used and for novel compounds NMR etc are used. HPLC is used as identification tool when standards are available and after spiking with the pure standard if a suspected peak sharply rises the compound is confirmed and if a shoulder is formed by that peak, then it is some other compound. Here we have used this spiking technique in HPLC analysis but as it is not so authentic technique therefore, its fraction has been subjected to GC-MS analysis which is an authentic technique. In short HPLC has been used for separation whereas GC-MS has been used for the identification the compounds present.
  • Also, we were interested in purified fraction rather than crude extract as crude extract contained a range of compounds for which then one is not sure which compound will be the responsible compound of the observed biological activity. If you use a purified fraction the selection become narrow and authentic.

Line 188- “specie” check spelling

  • Worthy reviewer, corrected accordingly.

Line 209-211- “Met + Ext and 7 purified fractions inhibited by DPPH revealed that F4, F6, and F7 caused significant inhibition with the lowest IC50 values 110, 105, and 110, respectively, comparable to ascorbic acid”- check the sentence construction.

  • Worthy reviewer, corrected accordingly.

Line 215-217- no reference cited

  • Worthy reviewer, sentence was deleted

Why the authors have not incorporated the antidiabetic potential in the discussion which was explored in the present study?

  • Relevant Discussion was incorporated accordingly.

Materials and Method

Line 221- could have added the latitude and longitude coordinates.

  • They were incorporated accordingly

Line 238- the modification made to the standard protocol needed to be explained.

  • Similar protocol followed as cited in manuscript. The “some modifications” words was removed from the sentence.

Line 241- it would be better to use antibacterial instead of antimicrobial as bacteria was only involved in the study.

  • Worthy reviewer, corrected accordingly.

Line 246- authors could include a note on various cultural media used

  • The required detail has been added in line 334 to 337 in the respective section accordingly

 Line 3- (title) It would be better to use antibacterial potential instead of antimicrobial potential because the study mainly focuses on various bacterial groups and in conclusion the authors have pointed out the same.

  • Worthy reviewer, antimicrobial was replaced by antibacterial as you suggested.

Round 2

Reviewer 1 Report

The authors have improved the manuscript significantly. I have no more comments.